# How Does the Experience Quality of Recreational Activities Organized within the Scope of Public Health Affect Perceived Value, Satisfaction and Behavioral Intentions of Individuals?

**DOI:** 10.3390/ijerph20065142

**Published:** 2023-03-15

**Authors:** Ozgur Yayla, Arif Aytekin, Fatih Uslu, Gozde Seval Ergun, Huseyin Keles, Yigit Guven

**Affiliations:** 1Department of Recreation Management, Manavgat Tourism Faculty, Akdeniz University, 07600 Antalya, Türkiye; 2Department of Social Work, Manavgat Social Sciences and Humanities Faculty, Akdeniz University, 07600 Antalya, Türkiye; 3Department of Educational Sciences, Education Faculty, Akdeniz University, 07600 Antalya, Türkiye; 4Department of Tourism Management, Manavgat Tourism Faculty, Akdeniz University, 07600 Antalya, Türkiye; 5Department of Tourism Guidance, Manavgat Tourism Faculty, Akdeniz University, 07600 Antalya, Türkiye; 6Department of Recreation Management, Graduate Education Institute, Ankara Hacı Bayram Veli University, 06830 Ankara, Turkey

**Keywords:** Experience quality, perceived value, satisfaction, behavioral intention, recreation

## Abstract

This study has been conducted in a sample of Eskişehir city center, which is one of the important destinations in Turkey, to determine the effects of experience quality in recreational activities on perceived value, satisfaction, and behavioral intention. Following this purpose, the data were obtained from 420 people who participated in recreational activities organized by the local governments between April and June 2022. As a result of the findings, it has been determined that the perceptions of individuals regarding the experience quality of the activities positively affect their perceptions of value and satisfaction. Moreover, individuals’ positive value perceptions of activities increase their satisfaction and behavioral intentions. This study differs from other studies by examining the variables of experience quality, perceived value, satisfaction, and behavioral intention in recreational activities as a whole. Most studies on recreation, which is accepted as important in the context of public health, in addition to the fact that recreational activities emphasize the perspective of protecting and improving health. Unlike other studies, this one is expected to contribute to the literature by determining the antecedents for the activities to be much more efficient.

## 1. Introduction

The industrial revolution brought along various problems (various physical health problems, mental health problems, the need for socialization and rest, etc.), while it also provided many benefits by directing humanity toward a different working order. It has become increasingly clear that humans cannot function like machines and that productivity can decrease if we only focus on eating, sleeping, and working every day [1]. It has also been understood that human beings need more than that; they need to relax, have fun, and engage in activities that bring joy and pleasure to their lives. Therefore, the importance of recreational activities in promoting public health, as well as the quality of experience in these activities, has been gaining more attention in recent years [2,3,4]. This situation highlights the need for countries and employers to address not only the economic needs of their citizens and employees but also their physical, psychological, and social needs. In particular, employers should recognize the importance of providing access to and promoting recreational activities for their employees as a means of enhancing employee well-being and productivity. Because factors such as job stress and challenging working conditions reduce the employees’ quality of life, they may be prompted to consider quitting their jobs [5]. Mansour and Tremblay [6] argue that the conflict regarding the time spent on work and leisure activities increases employee burnout and turnover. This result makes it essential for organizations to provide opportunities for recreational activities and to establish a balance between work and leisure. Recreation is a concept that can meet the needs of people both physically, mentally, and socially, and it can motivate them to meet those needs [7]. Additionally, recreational activities can help individuals relax, have fun, and socialize with others, which can improve their physical and mental health. It can also help them build social connections. In this regard, Litwiller et al. [8] analyzed the studies in the literature about the benefits of recreation. They found that recreational activities had a considerable effect on individuals’ ability to socialize and have a pleasant time. Furthermore, recreational activities can help individuals develop new skills, interests, and hobbies that can contribute to their personal and professional development [9,10]. In this way, individuals have both increased their performance in their working and social lives and become more productive. This has led to changes in how and what people do for fun, and fun has become a very important part of society [11].

As the importance of recreation has increased, it has also become clear how important it is for people to have more leisure time and to use it effectively. At this point, destination management organizations in city centers have organized many recreational activities to protect public health and designed city centers to accommodate these activities. This is because public parks or areas in the destination contribute to individual health and increase the livability of the region [12].

On the other hand, the efficient use of leisure time is related to the level of experience quality provided by the activity. Studies have shown that when individuals’ perceptions of experience quality are low, the value they place on the activity, the satisfaction they get from the activity, and their behavioral intentions are significantly affected [13]. Although the concept of experience quality is not widely discussed as a research topic in the recreation literature [13,14,15,16,17], it is a frequently studied topic in the fields of tourism [7,8,9,10,11,12,13,14], sports [18], and health [19,20].

The concept of experience quality has been extensively researched in recent years, especially in the field of tourism, but it does not have as much detailed and long-term research as the concept of service quality [21]. The perception of experience quality has been defined as the psychological and social impact of the action or event that individuals experience [13] and has been considered part of the service experience [21]. However, the scope of individuals’ perceptions of experience quality is broader than that of service quality perceptions because experience quality encompasses feelings and emotions that cannot be measured by service quality. Therefore, the perception of experience quality should be more frequently handled as a research topic in the fields of recreation, tourism, sports, and health.

The previous studies conducted on experience quality have indicated that it affects many different concepts. Research has shown that individuals’ perceptions of experience quality affect their satisfaction [22,23], their perception of value related to the action or event [24,25,26], and their behavioral intentions [27,28,29,30,31,32].

In this century, organizing activities in city centers, public parks, and recreational areas as a means of promoting public health is among the top priorities of all destinations [12]. However, the scarcity of studies on experience quality in the field of recreation has created a gap in the field. To this end, this study aims to fill that gap by measuring the perceptions of experience quality in recreational activities and examining whether this concept affects individuals’ perceived value, satisfaction, and behavioral intentions.

## 2. Conceptional Framework

### 2.1. Experience Quality and Perceived Value

The concept of experience quality has emerged as a component of service quality. Experience quality, which is a more subjective concept than service quality, also addresses the emotions and feelings that contribute to the overall quality of the service experience. Otto and Ritchie [33] examined the differences between service quality and experience quality in their study. They stated that the concept of service quality is more objective, while the concept of experience quality is more subjective. According to Otto and Ritchie [33], experience quality is more general and hedonic than utilitarian. In addition, they stated that the quality of experience has a holistic tendency rather than being quality-based [33]. 

In the literature, it is observed that the five-dimensional scale developed by Parasuraman et al. [34], which includes physical assets, reliability, eagerness, trust, and empathy, has been frequently preferred to measure service quality. Though service quality measurements have been made in different fields, it is known that the dimensions of service quality measured are similar. For example, MacKay and Crompton [35] used the five-dimensional scale developed by Parasuraman et al. [34] to measure service quality in recreational activities. McDonald et al. [36] also used the same scale for the evaluation of service quality in sports centers. As for measuring the experience quality, there is no measurement model on which researchers have reached a consensus. This is because experience is a memorable and multidimensional phenomenon that affects the consumer. That is why experience quality has been measured in different dimensions within different fields. For instance, Kao et al. (2008) proposed a four-factor structure including fun, participation, surprise, and immersion to measure the experience quality of individuals visiting theme parks. Additionally, Chen and Chen [13] mentioned a three-factor structure including involvement, peace of mind, and education to measure the experience quality of tourists visiting cultural heritage sites. Çetiner and Yaylı [37] examined the experience quality perceptions of individuals participating in animation activities through the dimensions of immersion, surprise, involvement, entertainment, and education. However, some studies measure the experience quality as a whole in a single dimension [38], as multidimensional and long scales may pose problems in the measurement of constructs having more than one dependent variable [39].

On the other hand, perceived value is associated with a relative comparison between the sacrifices individuals make (price, time, effort, and risk) and the benefits obtained from the products or services consumed [40]. Perceived value is based on equity theory. The difference between the consumer’s sacrifices and the benefit it provides refers to the individual’s perceived value [41]. If the ratio between individuals’ sacrifices and experiences is equivalent, the perceived value will be high [42,43]. If the participants and their service providers perceive that they benefit more than they expect in monetary and non-monetary terms, this situation positively affects their future behavior [44]. Based on this information, it is thought that the quality of the experience in recreational activities may also positively affect the individuals’ perceived value. Accordingly, the research hypothesis has been developed as follows:

**H_1_:** 
*The experience quality of recreational activities affects the perceived value of individuals.*


### 2.2. Experience Quality and Satisfaction

In the field of recreation, the service quality concept refers to service performance at the quality level. Experience quality, on the other hand, explains the psychological outcome that emerges as a result of the participation of individuals. Experience quality can be conceptualized as the emotional responses of individuals to the desired social-psychological benefits. When we examine the experience quality studies in the context of recreation, it is seen that the issue has been investigated in many ways. For example, there are studies in the literature that investigate how the experience quality of recreational climbing can be improved in intense weather conditions and how recreational climbing can be more fun and safe [16] and whether the quality of the experience can increase when the service performance threshold is exceeded in activities in the entertainment sector [14]. Moreover, the relationship between place perception and visitors’ perceptions of experience quality in natural parks containing wildlife [45,46], the effects of preferences in recreational activities on experience quality and satisfaction [17], and the mediating role of experience quality on authenticity and satisfaction in the context of cultural heritage tourism [47] are also among the topics examined.

On the other hand, satisfaction refers to the perceived inconsistency between consumers’ expectations and perceived performance after consumption [30]. Dissatisfaction arises when performance is different from expectations [30]. Satisfaction can be defined as the degree to which an experience evokes positive emotions [48]. In the context of recreation, satisfaction is primarily expressed as the difference between preferred pre-activity expectations and post-activity experiences [49]. In this direction, it can be said that as the experience quality of recreational activities increases, individual satisfaction will also increase. In light of this information, the research hypothesis has been developed as follows: 

**H_2_:** 
*The experience quality of recreational activities affects the satisfaction of individuals.*


### 2.3. Perceived Value and Satisfaction

When the studies on perceived value are examined, the most commonly researched themes are the relationship between customer experience and perceived value [41], the effect of experience quality and loyalty on perceived value [50], the relationship between motivation to volunteer and perceived value [51], the relationship between the price paid by spectators in combat sports and perceived value [52], perceived value in health tourism [20,53], the relationship between perceived value and behavioral intention in golf tourism [31], the relationship between perceived value, satisfaction, and behavioral intention in tourism [54,55], and the role of festival designs in consumers’ perceptions of value [56].

Perceived value is a subjective phenomenon in many ways: it varies between individuals [34,57] and at different times [58]. Moreover, demographic variables such as age, income level, and marital status affect the perceived value of individuals. This situation may cause differences in satisfaction levels [59]. It is important to investigate the effect of perceived value in recreational activities, which is an important element in terms of public health, on satisfaction in this context. Accordingly, the research hypothesis has been developed as follows:

**H_3_:** 
*The perceived value of recreational activities affects the satisfaction of individuals.*


### 2.4. Perceived Value and Behavioral Intention

Oliver [60] defines customer loyalty in four stages: cognitive loyalty, emotional loyalty, traditional loyalty, and action loyalty. In practice, it is difficult to measure action loyalty, and therefore most researchers have used behavioral intentions [61]. When we examine the studies in the literature on behavioral intentions, it can be observed that they have looked at the effect of satisfaction at festivals and events on behavioral intentions [62,63,64,65], the role of behavioral intentions in determining travel goals [66], the relationship between value and behavioral intentions in touristic shopping [67], and the relationship between brand awareness of slow cities and behavioral intentions [68] so far. On the other hand, the effect of perceived value in recreational activities on behavioral intention remains unclear. In this context, the research hypothesis has been developed as follows:

**H_4_:** 
*The perceived value of recreational activities affects the behavioral intentions of individuals.*


### 2.5. Satisfaction and Behavioral Intention

The concepts of perceived value and customer satisfaction are considered among the terms that have attracted a great deal of attention and have been studied in the field of recreation. The most important reason for this is that perceived value and customer satisfaction have significant effects both on the post-purchase thoughts and decisions of customers. Nevertheless, since perceived value and customer satisfaction are close concepts, they are often confused, and the nuances between them are overlooked [69].

Although perceived value is a cognitive evaluation that emerges before and after the purchase, customer satisfaction is an emotional outcome that arises after the purchase, and, beyond being cognitive, it can be effective in directing the future purchasing attitudes and tendencies of the customer [70]. Since perceived value is related to how customers evaluate what is offered to them, it requires a strategic view of how the business can best meet customer needs. However, customer satisfaction needs a tactical view of how well service delivery can be achieved [71]. While customer satisfaction is the result of the evaluations of existing customers, perceived value is the result of the evaluations of past, present, and potential customers [72]. Last but not least, while perceived value is the result of the customer’s judgment as a result of evaluating the products and services offered to him/her and competitor products and services, customer satisfaction results from the evaluation of only what is offered by the company [69,73].

Previous studies have shown that service quality and perceptions of value affect satisfaction, which in turn influences loyalty and behaviors [74,75]. In the literature on recreation, studies have looked at the relationship between tourist satisfaction [76,77,78,79,80,81,82,83], leisure activities, and life satisfaction [84,85]. Moreover, the relationship between the content of campus recreational activities and satisfaction [86] and the relationship between participant satisfaction and the desire to re-participate in sporting events [87] have also been examined.

Positive behavioral intentions generally represent consumer loyalty. Customer loyalty is a key component for the long-term viability or sustainability of a business and is therefore a particularly important topic for businesses in the marketing literature. Measuring loyalty can help better understand how to retain customers. Retaining existing customers generally requires lower costs than acquiring new ones. Moreover, loyal customers are more likely to act as word-of-mouth agents and recommend a product/service to their friends, relatives, or other potential customers [88]. Based on this information, it is thought that satisfaction with recreational activities plays a leading role in the formation of behavioral intentions. Therefore, the research hypothesis that has been formulated is: 

**H_5_:** 
*Satisfaction with recreational activities affects the behavioral intentions of individuals.*


## 3. Materials and Methods

### 3.1. Research Instrument 

In this study, the effects of individuals’ perceptions regarding experience quality in recreational activities on value, satisfaction, and behavioral intentions have been examined. Experience quality was measured by eleven statements defined by Wu et al. [89]. Perceived value was measured by six statements adopted by Kim et al. [90]. To measure satisfaction, five items were used based on existing literature [89,91]. To measure behavioral intentions, three items were adopted from previous research [92,93,94,95]. All these items were measured using a 7-point Likert-type scale ranging from seven (strongly agree) to one (strongly disagree).

### 3.2. Sampling and Data Collection

The research was carried out in Eskişehir, one of the most important urban centers in Turkey. In Eskişehir city center, there are public centers, walking paths, museums, cultural and artistic events, festivals, art galleries, cinema screenings, small or large-scale parks, and areas offering sporting activity facilities and opportunities to meet the daily needs of people [96]. Eskişehir hosts many sporting and artistic activities and events, especially those that contribute to public health. Approximately 600,000 people live in the city center and according to the 2022 statistics of Eskişehir Metropolitan Municipality, the number of individuals benefiting from recreational activities in the province was determined to be 2,868,716 [96]. On the other hand, Eskişehir, which was chosen as the Cultural Capital of the Turkish World in 2013 and 2014, was chosen as the study universe due to the increase in recreation areas. 

Before starting the data collection process, in January 2022, a pilot study was conducted on 20 people. As a result of the pilot study, the reliability values of the scales used were examined, and it was determined that the Cronbach alpha value of each scale was at least 0.814. This value is above the value suggested by the literature [97]. In addition, as there was no problem with the clarity of the questions as a result of the pilot study, the stage of collecting the actual research data was started. The questionnaire was applied to 445 people aged 18 and over who participated in the activities organized by the local authority in Eskişehir city center between April and June 2022. After excluding the erroneously completed or incomplete forms, the research data collection process was completed with 420 questionnaires. 

### 3.3. Common Method Bias

Since all constructs have been measured using the same methodology, the risk of common method bias is high in the social sciences [98], and several procedures have been applied. For each questionnaire form, a cover page was prepared with information such as “Participation is optional,” “The data collected during this research will be kept confidential,” and “There is no right or wrong answer in this survey” [99]. Regarding statistical solutions, Harman’s single-factor test was applied. After entering all variables into an exploratory factor analysis (EFA), the unrotated factor solution revealed that no single element accounted for the majority of the variance (the largest identified factor explained 19.4% of the variance).

### 3.4. Data Analysis

First of all, the data collected from the research was transferred to the SPSS Statistics Base V23 program. Before the structural model examination phase, the data screening process was applied. As part of the data screening process, the first extreme values were examined and the Mahalanobis distance was calculated. At this point, eleven questionnaire forms with extreme values were excluded from the analysis (Mahalanobis’ D (25) > 0.001). The effects between the relevant variables were determined using the remaining 409 questionnaires. In the second stage of the process, the data were examined to determine whether there was a multicollinearity problem or not. In this context, it has been determined that the VIF value of each scale is at its maximum of 1.206 and the tolerance value is at its minimum of 0.544. Since the values were in the desired range, it was decided that there was no multicollinearity problem [100]. In the last stage, the Skewness and Kurtosis values of the data were examined, and it was determined that their values were between −1.5–+1.5. These results indicate that the data show a normal distribution [101]. 

The AMOS program was used to test the structural model, which was developed depending on the purpose of the research. In this context, the two-stage approach, suggested in the literature, was preferred [102]. First, the data were subjected to confirmatory factor analysis. Based on the satisfactory results obtained, path analysis was performed, and hypotheses were tested.

## 4. Results 

### 4.1. Demographic Profile

When the table has been examined, it is understood that 71.4% of the participants are female and 28.6% are male. According to the data in the table, about 60% of the participants are between the ages of 25 and 45. In addition, approximately 70.0% of participants are married. Finally, when the education levels are examined, it is seen that 59.7% of the participants are undergraduate graduates. Data on demographic characteristics are given in Table 1.

### 4.2. Confirmatory Factor Analysis Regarding the Structural Model

The fits of the scales were assessed using confirmatory factor analysis (CFA). The structural equation modeling (SEM) procedure was performed to test the relationships between the four constructs in the model. One exogenous variable (experience quality) and three endogenous variables (perceived value, satisfaction, and behavioral intentions) were tested to determine how well the proposed model fit the data. AMOS structural equation modeling (SEM) was employed with the Maximum Likelihood (ML) method of parameter estimation.

Each construct was analyzed with CFA to confirm the measurement scale property in the model. Hair et al. [103] stated that the ideal value for factor loads is 0.70 and above. For this reason, as a result of the analysis made, two of the experience quality scales (I believe that the activity I participated in provided me with an educational and guiding experience; the qualities of this activity are at a superior level compared to other organized activities); perceived value (generally, it was a good action to participate in this activity); and satisfaction (services provided in these activities were generally satisfactory) factor loads of one expression were found to be below 0.70 and were excluded from the analysis. The CFA was applied to the four-factor structure with twenty-one expressions, and the results are shown in Table 2. When the table is examined, it is seen that factor loads vary between 0.749 and 0.965. The results showed that the measurement model of this study demonstrated, overall, a high degree of goodness of fit to the data: x2/df = 3.349, RMSEA = 0.076, GFI = 0.88, CFI = 0.96, AGFI = 0.84, and RMR = 0.075.

It has been suggested in the studies that the composite reliability value should be above 0.70 [103,104], and the average variance extracted value should be above 0.50 [103]. It has been determined that these two aforementioned values are higher than the recommended limit values. Therefore, it can be stated that the construct reliability and convergent validity of the quadruple model are appropriate.

### 4.3. Hypothesis Testing

With the maximum likelihood method, a structural equation model was used to investigate relationships among the four constructs in this model. The results of maximum likelihood estimation have provided an adequate fit to the data: x2/df = 3.444, RMSEA = 0.077, GFI = 0.91, CFI = 0.97, AGFI = 0.87, and RMR = 0.087. 

Path analysis results are shown in Table 3. When the table has been examined, it is understood that the experience quality has a strong effect on the perceived value (t = 16.71, *p* < 0.001), therefore H_1_ has been supported. H_2_ has predicted that the higher the experience quality, the higher the level of satisfaction, and in this sense, H_2_ has also been supported (t = 4.71, *p* < 0.001). Hypotheses 3 and 4 have been supported by showing that perceived value is a significant predictor of satisfaction (t = 7.34, *p* < 0.001) and behavioral intention (t = 3.46, *p* < 0.001). Finally, it is understood that satisfaction has a positive effect on behavioral intention, so H_5_ has been supported (t = 5.16, *p* < 0.001). The ultimate results of the analysis have been summarized in Figure 1.

## 5. Discussion and Implications

This study aimed to investigate the effects of the quality of the experience of individuals participating in recreation activities on the perceived value of the activity and their satisfaction and behavioral intentions. In addition, it is to determine whether the concept of perceived value affects people’s satisfaction with the activity and their behavioral intentions. 

The importance of experience quality in the research findings has been determined to be such that it directly affects the perceived value of individuals and is an antecedent of satisfaction. This finding is consistent with Yuan and Wu’s [105] assertion that experiential quality is defined as an important indicator of perceived value.

This situation has shown that if the activities are conveyed to the individuals correctly and a quality service is provided, the perceived value of the participants will be positively affected. These results are in line with and supported by other studies in the literature. Petrick et al. [106] stated that if the experience quality in recreational activities is improved, the perceived value of individuals is positively affected. Perera and Vlosky [107] stated in their research that increasing the travel quality would have a positive effect on the perceived value of individuals and also have an effect on their behavioral intentions. 

In addition, this study is parallel to the study conducted by Chen and Chen [13], which found that there is a significant relationship between quality of experience and satisfaction. Based on the findings of the study, it has been recommended that institutions and organizations, especially those engaged in recreational activities, should plan their activities by considering the quality of experience of the individuals participating in all stages of their formation. It can be stated as an important issue since an activity in which experience quality is given importance will have a positive effect on the participants’ perceived values. At this point, local governments can work to improve the landscapes of recreation areas to increase the quality of the experience for individuals. Similarly, employing the necessary number of personnel in the areas where activities are organized and coordinating each activity by recreation leaders who are experts in their fields will increase experience quality.

It has been determined that if recreation activities are implemented correctly, they affect the experience quality of the participants, and this situation creates satisfaction in the participants. Hypothesis H_2_ suggests that as the quality of experience increases, the satisfaction of individuals with the activity increases. In addition, Fernandez et al. [41] examined the perception of experience quality in low-cost sports centers in their research and stated that as the experience quality of individuals participating in activities increases, satisfaction and behavioral intention are positively affected. Suhartanto et al. [21] and Prebensen, Kim, and Uysal [108] also found similar results. In 2016, Prebensen, Kim, and Uysal [108] emphasized that the perceptions of the quality of the experience of individuals participating in winter tourism activities have a positive effect on their satisfaction. In addition, Pramod and Nayak [109], in their study in 2018, stated that the emotional experiences of tourists have a positive effect on satisfaction and presented data supporting the findings of our study.

The results show that experience quality has a positive effect on both the perceived value and satisfaction of individuals. For organization planners, satisfaction is the main goal to be achieved. On the way to reaching the main goal, the positive effect of experience quality should be carefully examined by the planners. Otherwise, it is thought that individuals’ satisfaction with the activity will decrease, and they will have negative feelings toward the organization, event, and activity. In order not to cause this, it is of great importance for the organizers of recreation activities to measure the quality of the experience they provide and to control this parameter in terms of the sustainability of the activity. 

As a consequence of the H_3_ testing, it has been found that as the perceived value of the individuals regarding activities increases, the satisfaction received from the activities also increases linearly. This is considered a serious result in expressing the importance of the concept of value in recreation literature. For this reason, the concept of satisfaction, which is frequently studied, especially in public health issues, also positively affects the activity outputs of individuals. In cases where participation in recreational activities is satisfying, it has also been observed that it can mediate the stress response and increase the quality of life of individuals [110]. For example, satisfying recreational activities have been shown to positively affect depression and anxiety symptoms in another study [111]. 

The concept of value perceived by the masses has an important place among the conditions for participating in recreational activities, which are known to positively affect the psychological, physical, and social well-being of individuals. The perception of the value of the activity is one of the reasons for preferring recreational activities involving individuals of all age categories. Since this situation directly affects the feeling of satisfaction, it has an important place in recreational activity planning. The reason why audiences prefer activities that they think will satisfy them is related to which activity they perceive to be more valuable. In this sense, it is thought that in every recreation plan, it is necessary to pay attention to the quality of the experience to increase the perceived value, which will create a situation that will bring satisfaction. 

In another finding of the study, it has been stated that the concept of perceived value positively expresses behavioral intention through the H_4_ hypothesis. The fact that individuals’ behavioral intentions regarding their participation in recreational activities are positively affected means that their participation in the activity again will be positively affected. Increasing participation in recreational activities brings many benefits to individuals. It has been stated that increasing participation in recreational activities has a positive effect on individuals’ mental health, social and emotional states, and body composition [112]. As a result of the research, it has been observed through the studies conducted in the literature that the increase in behavioral intentions towards recreation activities increases participation and brings various social health benefits [113]. 

When the concept of behavioral intention is positive, it is thought to have a direct impact on the sustainability of the activity. On the other hand, an individual with a positive behavioral intention is stated to be highly likely to participate and share that activity again with the current research. Thus, the positive change in behavioral intention will increase the willingness of individuals involved in the activity to participate in new activities. The enthusiastic individuals will be motivated to share their positive feelings with others and to include them in the activities. This will trigger the collective participation intention and create a multiplied benefit in terms of public health. With these data, measurements should be made for the behavioral intentions of individuals in terms of the sustainability of activities. Every organization or recreation plan has to be sustainable due to the necessity of providing regular benefits. In this case, it is thought that the measurement of behavioral intentions is an important phenomenon.

Finally, using hypothesis H_5_, it has been determined that the increase in satisfaction with activities has a positive effect on behavioral intention. There are studies in the literature that are in parallel with this result of the research [13,21,55,114]. In the aforementioned studies, it has been determined that satisfaction with activities has a positive effect on behavioral intentions. It has been observed that if individuals think that the activities they participate in are worth the effort, time, and money they spend, their satisfaction will increase, and in addition, when this value is high, they will be more likely to participate in the activity in the future. 

## 6. Limitations and Future Research Directions

In addition to its significant contributions to the literature, the current study has several limitations. The first limitation is that the study has been conducted at a single destination. Therefore, to determine the external validity of the results of the study, similar studies in other countries and regions will be instructive for future research. 

The concept of quality of experience examined in the research is particularly effective in the participation of recreational activities, in other words, in being preferred again. Increased participation is thought to facilitate the emergence of benefits arising from recreational activities. On the other hand, considering the benefits of recreational activities, it is stated in the literature that they positively affect public health and community well-being. 

In future studies, the antecedents of recreation experience quality can be investigated. For example, the effect of motivation on experience quality can be analyzed. In addition, it is thought that variables such as age and gender may show differences in experience-quality perceptions. In addition, the economic status of the participants, their working hours, or whether they have any disabilities may differentiate their perceptions of the quality of their experiences in recreational activities and the effects of this perception on other dependent variables. In this context, it is recommended to evaluate whether individuals with different demographic characteristics differ in the quality of experience in recreational activities in future studies. 

## 7. Conclusions

The structural model developed through this study has examined the behaviors of the participants by exploring the causal relationships between activity quality, perceived value, satisfaction, and behavioral intentions. The interrelationships between the concepts of experience quality, satisfaction, and behavioral, intentions have been examined, and the findings have important implications for recreation managers and organizers.

Improving the quality of recreational activities, creating the necessary infrastructure, providing various pieces of training, and choosing qualified individuals are among the factors that should be taken into consideration by practitioners. Moreover, the managers who implement recreational activities should keep up with current information and constantly look for innovative ways.

In addition to the points mentioned earlier, it is also important to note that experience quality can be influenced by a variety of factors, including the physical setting, the nature of the activity, and the social interactions that occur during the activity. Understanding how these factors contribute to experience quality can help in the design and management of recreational spaces and activities. Another important aspect to consider is that different individuals may have different perceptions and evaluations of the quality of their experiences. For example, some individuals may prioritize adventure and thrill-seeking, while others may prioritize relaxation and tranquility. Understanding these individual differences and how they influence experience quality can help in tailoring recreational activities and spaces to better meet the needs and preferences of different people.

Furthermore, it is also important to note that experience quality has a significant impact on an individual’s overall well-being and quality of life. Studies have shown that participation in recreational activities can have a positive impact on physical health, mental health, and social well-being. This is particularly important given the increased importance of recreational activities in the context of modern society, where people often experience high levels of stress and pressure.

In conclusion, this study aims to contribute to the literature by providing new insights into the relationship between experience quality and behavioral intention in the context of recreational activities It also has the purpose of helping us better understand the factors that influence experience quality and how they can be used to improve overall well-being. The findings of this study can be used to inform the design and management of recreational spaces and activities, as well as the tailoring of recreational activities to meet the needs and preferences of different individuals.

## Figures and Tables

**Figure 1 ijerph-20-05142-f001:**
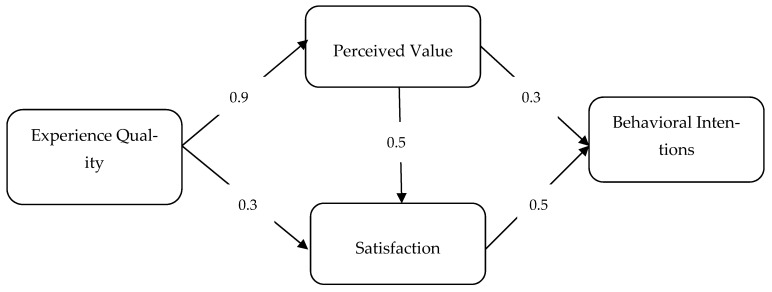
Hypothetical Model Testing Results.

**Table 1 ijerph-20-05142-t001:** Description of the Participants (*n* = 409).

Variables	Frequency	Percentage (%)
Gender		
Male	117	28.6
Female	292	71.4
Age		
18–24	66	16.1
25–34	118	28.9
35–44	117	28.6
45–54	88	21.5
55 or more	20	4.9
Marital Status		
Married	285	69.7
Single	124	30.3
Education level		
Primary education	39	9.5
Secondary education	102	24.9
Graduate	244	59.7
Postgraduate	24	5.9

**Table 2 ijerph-20-05142-t002:** Overall CFA for the measurement model (*n* = 409).

Construct and Indicators	Factor Loading	t-Value	C.R.	AVE
**Experience Quality**			**0.952**	**0.690**
The quality of interaction between me and the staff working at the events is unmatched.	0.767	18.20 *		
The interaction between me and the staff working at the events is of a high standard.	0.751	17.64 *		
The physical environment in which these activities are held is perfect.	0.756	17.81 *		
The physical environment at these events is of a high standard.	0.749	17.56 *		
The service provided by the personnel working at the events to the participants of the events is of high quality.	0.849	21.20 *		
Every time I attend events, I have a unique experience.	0.847	21.05 *		
I feel free to participate in this activity.	0.851	21.29 *		
This place, where I participate in activities, is at an accessible point.	0.772	18.27 *		
Participating in this activity is a good experience for me.	0.826	-		
**Perceived Value**			**0.957**	**0.819**
I think I get a higher benefit from the activity compared to the price I pay.	0.832	-		
I think I get a higher benefit from the activity compared to the effort I put in.	0.857	32.10 *		
I think I get a higher benefit from the activity compared to the time I spend.	0.872	22.33 *		
Although this activity was free or very low cost, it met my needs.	0.881	22.73 *		
Overall, it’s worth the time, money, and effort to participate in this activity.	0.891	23.19 *		
**Satisfaction**			**0.956**	**0.846**
I am satisfied with my experience as a result of participating in this activity.	0.889	-		
I am happy that I decided to participate in these activities.	0.934	30.40 *		
I really enjoyed participating in these activities.	0.927	29.80 *		
Overall, I felt energized after participating in this activity.	0.828	23.13 *		
**Behavioral Intention**			**0.917**	**0.789**
I would like to attend these events again in the future.	0.923	33.45 *		
I will recommend participating in these events to my friends and close circle.	0.965	18.11 *		
I am willing to pay more to participate in these activities.	0.765	-		

* *p* < 0.001.

**Table 3 ijerph-20-05142-t003:** Results of the proposed model.

Hypothesized Path	Standardized Coefficients	t-Value
H_1_: experience quality → perceived value	0.91	16.71 *
H_2_: experience quality → satisfaction	0.36	4.71 *
H_3_: perceived value → satisfaction	0.58	7.34 *
H_4_: perceived value → behavioral intentions	0.34	3.46 *
H_5_: satisfaction → behavioral intentions	0.51	5.16 *

* *p* < 0.001.

## Data Availability

The data analyzed during this study are available on request from the corresponding author.

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
