# Peer review of "How Does the Experience Quality of Recreational Activities Organized within the Scope of Public Health Affect Perceived Value, Satisfaction and Behavioral Intentions of Individuals?"

_ijerph, 2023, doi:10.3390/ijerph20065142_

Round 1

Reviewer 1 Report (Previous Reviewer 1)

Review 2

These are my comments in relation to parts of the paper:

Abstract

The conclusion is hard to follow. It is not usual to write in the conclusion who will use the research results, rather write a clear scientific or practical contribution.

Introduction

In the introductory part, it would be desirable to take into account at least two or three recent articles from the last 5 years on the importance of recreational activities for public health. The cited literature is of a much older date, and in this way the actuality of the topic would be emphasized.

Materials and Methods

Line 209 – „Experience quality was measured by eleven statements defined by [71]” –  end the sentence with what is defined, and leave the reference

Line 210 – “Perceived value was measured by six statements adopted from [72]” - finish the sentence on the basis of which it was adopted, and leave the reference

Results

The results are clearly presented.

Discussion and implications

The discussion has been correctly and interestingly written. Necessary references have been added.

Limitations and Future Research Directions

Do the authors think that in some future research it would be useful to take into account data on employment, economic status, health status, the presence of disability of the respondents, etc. within the description of the sample?

Conclusions

Line 494-497 – “Besides, the findings of the research have shown that it can be beneficial for recreation managers and practitioners …” - the research did not look at the benefits for managers and organizers involved in planning recreational activities, and it cannot be claimed that the research showed that. In conclusion, stand by your results, and further research is needed to prove it.

Author Response

Thank you so much for your positive comments and assessment of our topic. Your comments and suggestions have been very helpful for further improving of our paper.

We greatly appreciate your feedback.

The changes which were made about the manuscript have been highlighted in our text.

Reviewer 2 Report (Previous Reviewer 2)

Though you have improved on the previous manuscript, alot of issues remain outstanding. The abstract, background, discussion and conclusion in particular are disjointed and incoherent. A number of comments are given intext and could assist in improving the manuscript.

Author Response

Thank you so much for your positive comments and assessment of our topic. Your comments and suggestions have been very helpful for further improving of our paper.

We greatly appreciate your feedback.

The changes which were made about the manuscript have been highlighted in our text.

This manuscript is a resubmission of an earlier submission. The following is a list of the peer review reports and author responses from that submission.

Round 1

Reviewer 1 Report

Although the authors state that the topic of their work is related to public health, the content of the work is more suitable for another area, such as sports, recreation, environment, tourism, etc.

The abstract is not written sufficiently informative. Sample data and results are missing.

Within the introduction and discussion, references are missing in numerous places.

The authors conducted the research with a self-made questionnaire, combining questions from similar questionnaires.

409 respondents participated in the research. In the description of the sample, only data on the gender, age and marital status of the respondents are given. There is a lack of data on the educational structure and employment of the respondents, which would give a more complete picture of the sample

In the statistical analysis, it would be preferable to check the normality of the distribution with the help of the Kolmogorov-Smirnov test and multivariate normality.

Reviewer 2 Report

The manuscript “How Does The Experience Quality of Recreational ………….Value, Satisfaction and Behavioral Intentions” addresses an important gap in literature and can be further improved based on the following comments.

General comments

There is need to pay attention to the organisation of the manuscript, coherent synthesis of the main findings and flow of the manuscript. Elaborate comments are given in text.

Specific comments

 Title: Some of the words in the title can be deleted 

 Abstract

Need to be revised as per in text comments especially with regard to background of the study.

Organisation

·       Cases of disjointed, repetition and in appropriate tenses (see highlighted text/ words) for example the reader may fail to understand whether you are addressing health or recreation.

·        Many of the tests as indicated in text are placed in wrong sections.

Method

·       Need to improve coherence of this section. For example, under study area, adequate description of the study area should be given. 

·       Description of statistical methods and processing of data is disjointed.

·       What are the sampling strategy and its justification and how did you arrive at the sample size?

Discussion

·       The discussion is too lengthy and lacks analysis and synthesis.

·       Synthesis of the discussion section should be revisited as to make it clear, concise and have a logical flow without necessarily lifting statements from previous sections.

·       Failure to Summarise the main findings, considering the review’s objective(s), research question(s) and focus.

·       Strengths, limitations, and future research directions though present to an extent were not clear.

·       Where applicable, there is need to concisely compare and contrast your findings with the existing literature instead of lifting statements from other sections.

·       A clear line of reasoning is needed to link the findings (results section) with the implications (discussion and/or conclusion).

·       To that extend ensure that the description and explanation provided are coherent and provide plausible inferences

·       List the main implications of the findings and place these in the context of other relevant Literature instead of veering off
